# Infusing theory into deep learning for interpretable reactivity prediction

Shih-Han Wang[1,2], Hemanth Somarajan Pillai[1,2], Siwen Wang[1], Luke E. K. Achenie[1] & Hongliang Xin [1✉]

Despite recent advances of data acquisition and algorithms development, machine learning (ML) faces tremendous challenges to being adopted in practical catalyst design, largely due to its limited generalizability and poor explainability. Herein, we develop a theory-infused neural network (TinNet) approach that integrates deep learning algorithms with the well-established *d*-band theory of chemisorption for reactivity prediction of transition-metal surfaces. With simple adsorbates (e.g., *OH, *O, and *N) at active site ensembles as representative descriptor species, we demonstrate that the TinNet is on par with purely data-driven ML methods in prediction performance while being inherently interpretable. Incorporation of scientific knowledge of physical interactions into learning from data sheds further light on the nature of chemical bonding and opens up new avenues for ML discovery of novel motifs with desired catalytic properties.

[1] Department of Chemical Engineering, Virginia Polytechnic Institute and State University, Blacksburg, VA, USA. [2]These authors contributed equally: Shih-Han Wang, Hemanth Somarajan Pillai. ✉email: hxin@vt.edu

Adsorption energies of simple molecules or their fragments at solid surfaces often serve as reactivity descriptors in heterogeneous catalysis[1]. Rapid discovery of structural motifs with kinetics-favorable descriptor values, for example using quantum-chemical calculations, is appealing while remaining a daunting task due to the formidable computational cost in accurately solving the many-electron Schrödinger equation. In this aspect, the $d$-band theory of chemisorption pioneered by Hammer and Nørskov[2–6] has been widely used for understanding reactivity trends of $d$-block metals[7,8] and, to some extent, their compounds[9]. However, its quantitative prediction accuracy using individual $d$-band characteristics, e.g., the number of $d$-electrons[10], $d$-band center[2], and $d$-band upper edge[6,11], is limited due to the perturbative nature of the theoretical framework[12] and a large variation of site properties in high-throughput catalyst screening.

In recent years, machine learning (ML) has emerged as an alternative approach to predicting the chemical reactivity of catalytic sites with either hand-crafted[13–20] or algorithm-derived features[21–25]. By learning correlated interactions of atoms, ions, or molecules with a substrate from a sufficient amount of ab initio data, it is possible to compute adsorption properties orders of magnitude faster than traditional practices and narrow down candidate materials prior to experimental tests[13,14,16–18,22,25–28]. A major limitation of black-box ML models, particularly with the resurgent deep learning algorithms[29], is that it is easy to learn some correlates that look deceptively good on both training and test samples, but do not generalize well outside the labeled data. To alleviate the issue, active learning workflows guided by key performance indicators[17,30] and/or model uncertainties[16] have been used to accelerate the exploration of the enormous, essentially infinite, size of the accessible design space. Nevertheless, the necessity of a very large amount of data samples for model development and difficulties in interpreting model prediction impose tremendous challenges toward its adoption for automated search of high-performance catalytic materials.

Herein, we present a theory-infused neural network (TinNet) approach to predicting chemical reactivity of transition-metal surfaces and, more importantly, to extracting physical insights into the nature of chemical bonding that can be translated into catalyst design strategies. Incorporation of scientific knowledge of physical interactions into data-driven ML methods is an emerging area of research in catalysis science[13,18,19,23,24,31,32]. To the best of our knowledge, no such hybrid surrogate models of chemisorption were developed within a fully integrated ML framework that is reasonably accurate (~0.1−0.2 eV error) and transferable across diverse samples. By learning from ab initio adsorption properties with deep learning algorithms, e.g., convolutional neural networks, while respecting the well-established $d$-band theory of chemisorption in architecture design, the TinNet can be applied for a broad range of $d$-block metal sites and naturally encodes physical aspects of bonding interactions, inheriting the merits of both worlds. We demonstrate the approach using adsorbed hydroxyl (*OH) at {111}-terminated intermetallics and near-surface alloys as a representative descriptor species, such as in finding efficient electrocatalysts for metal-catalyzed $O_2$ reduction[33], $CO_2$ reduction[34], and $H_2$ oxidation in alkaline electrolytes[35]. This framework can be straightforwardly applied to other adsorbates (e.g., *O) or active site ensembles of multiple bonding atoms as shown for *N adsorption at {100}-terminated metal surfaces. The TinNet not only achieves prediction performance on par with purely regression-based ML methods, especially for out-of-sample systems with unseen structural and electronic features but also enables physical interpretation, paving the path toward ML discovery of novel motifs with desired catalytic properties.

## Results

**Deep network architecture.** As illustrated in Fig. 1, the TinNet framework contains two sequential components: a regression module and a theory module. The input into the regression module built with convolutional neural networks is the feature representation of the adsorbate–substrate system that encodes the atomic information and bonding interactions of each atom with its neighboring environment. The output units from the regression module then serve as unknown parameters in the theory module that is built upon the $d$-band theory of chemisorption for predicting adsorption properties of a $d$-metal site. To ensure model transferability, easily accessible graph features were used (see Fig. 1). In the graph-level scheme, each atom or node is represented by a binary vector, comprising nine properties of the atom, e.g., electron affinity, atomic volume, and electronegativity[26,36]. Similarly, each connection or edge encodes the pair interaction between neighboring atoms, including the solid angles swept out by the shared face of Voronoi polyhedra[22] and the kernelized distances[36]. A surface at the optimized bulk geometry with the adsorbate attached to the site of interest is used[37], thus avoiding the time-consuming structural optimization in the exploration of new systems[22]. Neural nets with $m$ convolution-pooling layers are connected to the feature representation sub-module. Within the convolutional layers, multi-dimensional feature arrays are iteratively updated by convolution (i.e., feature mapping) to extract high-level patterns and by pooling for feature subsampling. The 2D array is flattened into a vector, which can be fed into a fully connected network with $k$ hidden layers and a certain number of hidden neurons at each layer to capture the complex mapping between the extracted features and output targets. Finally, the output vector from the regression is incorporated into the theory module as local parameters along with user-defined global parameters, if any, that are independent of input features.

The physical meaning of each output unit from the regression module is pre-assigned in the TinNet framework. Historically, many factors have been used to correlate with the chemical reactivity of $d$-block metals, e.g., atomic or bulk properties[10,38], coordination numbers[39,40], and $d$-band characteristics[2,6]. Mapping physically relevant factors onto adsorption energies with ML algorithms has been previously explored with some success[13–15,17–19,21,25,31,32,41]. Besides the ambiguity of physical interpretation inherent to highly non-linear regression techniques, another major criticism is that some of the hand-crafted features require fully optimized geometric and/or electronic structures of the clean adsorption site, adding computational overhead costs to reactivity prediction of new materials. Instead of purely mathematical regression, we resort to the $d$-band theory of chemisorption with Newns–Anderson-type Hamiltonians[31,42,43] for computing the adsorption properties of metal sites. The central idea of the approach is to employ the activation output from the regression module as unknown, albeit trainable, parameters in the theory module (see Fig. 1). According to the $d$-band theory of chemisorption, chemical bonding at transition-metal surfaces can be conceptually separated into two consecutive steps[2]. First, the gas-phase adsorbate species, characterized by an orbital $|a\rangle$ at $\epsilon_a^0$, is embedded into the delocalized $sp$-states of the substrate, leading to a resonance state at $\epsilon_a$ with a Lorentzian line shape. Second, the adsorbate resonance interacts with a distribution of localized $d$-states $\rho_d$, shifting up in energies due to the orbital orthogonalization penalty for satisfying the Pauli exclusion principle (termed Pauli repulsion) and then hybridizing into bonding and antibonding states. The first step interaction with the $sp$-band contributes the largest part of chemical bonding, albeit as a constant $\Delta E_0$ for a given adsorbate and site type. The adsorption energy difference from one metal to the next is governed by the 2nd step $\Delta E_d$, which consists of Pauli repulsion and orbital hybridization[44], as

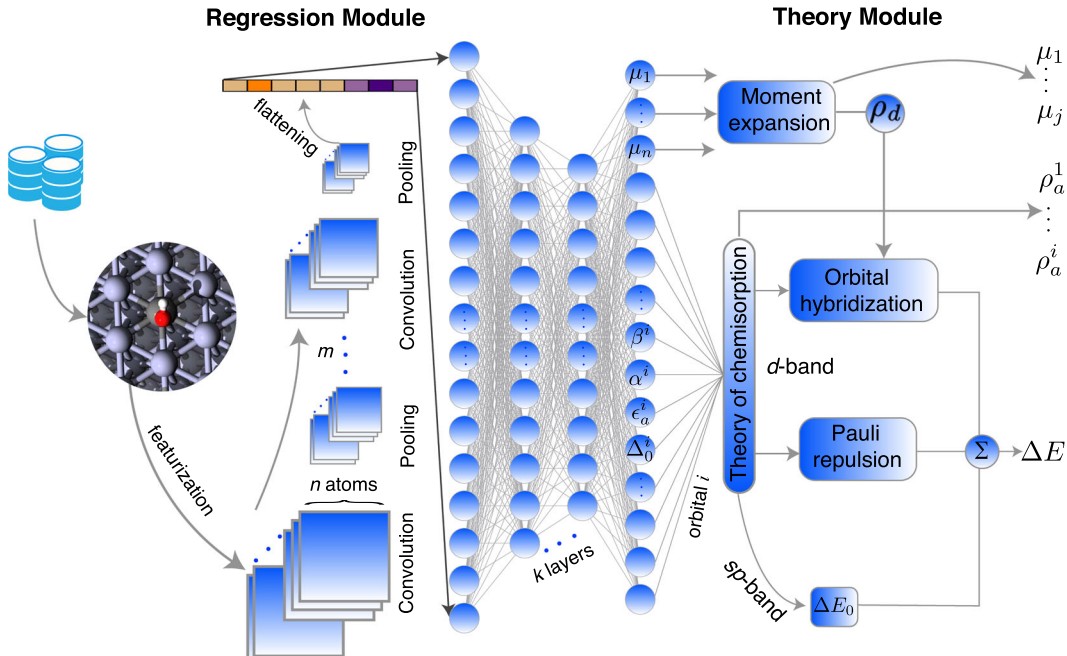

**Fig. 1 Schematic illustration of the theory-infused neural network (TinNet).** The information flows from the graph representation of a given adsorbate–substrate system to the adsorption energy $\Delta E$, the projected density of states onto the adsorbate frontier orbital(s) $\rho_a^1 \cdots \rho_a^i$, and $d$-band moments $\mu_1 \cdots \mu_j$ of the adsorption site. Circles and squares in the regression module represent neurons and feature maps, respectively.

illustrated in Fig. 1. The orthogonalization cost of interacting orbitals $\Delta E_d^{orth}$ can be quantified simply as proportional to the coupling integral $V$ and overlap integral $S$, which are related through $S \approx \alpha|V|$ ($\alpha$ as the overlap coefficient)[44]. $V^2$ can be conveniently written as $\beta V_{ad}^2$, in which $\beta$ denotes the coupling coefficient. $V_{ad}^2$ represents the interatomic coupling integral squared when the atoms are aligned along the $z$-axis and its standard value for a $d$-metal relative to Cu has been estimated from the linear muffin-tin orbitals (LMTO) theory and is readily available on the solid-state table[45]. To a first approximation, the $d$-band hybridization contribution $\Delta E_d^{hyb}$ can be obtained from one-electron eigenenergies using Green's function approach[43] with the parameterized Hamiltonian and the density of $d$-states $\rho_d$ as the input. The total adsorption energy $\Delta E$ is the sum of the energy contributions from the metal $sp$-states and $d$-states, $\Delta E_0$ and $\Delta E_d$, respectively. Another important piece of information from the $d$-band theory with the Newns–Anderson model is the density of states projected onto the adsorbate orbital $\rho_a$. Inclusion of multiple frontier orbitals $1 \cdots i$ of an adsorbate while considering their degeneracies can be realized by stacking fully connected network sub-modules (see Fig. 1). A full account of the theoretical framework was recently presented to bridge the complexity of electronic descriptors in understanding reactivity trends of pristine transition-metal surfaces and their alloys[31].

A TinNet model using the architecture in Fig. 1 can be considered as a complex function mapping the graph feature representation of an adsorbate-substrate system to adsorption properties, i.e., the adsorption energy $\Delta E$, projected density of states onto the adsorbate frontier orbital(s) $\rho_a^1 \cdots \rho_a^i$, and $d$-band moments $\mu_1 \cdots \mu_j$ of the adsorption site. Such mapping is parameterized by learnable weights of convolutional filters and neural connections in the regression module that is subsequently regularized by the theory module. The training of TinNet models can be performed by minimizing the sum-of-squares error loss function $J$ between model-predicted properties and DFT-calculated ground truths in the output layer (see Fig. 1). In the current TinNet implementation, two low-order moments ($\mu_1$, $\mu_2$) are embedded in the network for constructing the semi-ellipse $\rho_d$,

which is centered at $\epsilon_d$ ($\mu_1$, the first moment of the distribution relative to the Fermi level) with a full-width $W_d$ ($4\sqrt{\mu_2}$, $\mu_2$ is the second moment of the distribution relative to the center). This simplified distribution is sufficient in computing orbital hybridization energies compared with self-consistent, DFT-calculated density of $d$-states for transition-metal surfaces[11]. Higher-order moments of a distribution can be included using moment methods if necessary[6,46]. Using the backpropagation and stochastic gradient descent (SGD) algorithms, the constrained optimization can be performed. The PyTorch framework is used for implementing the hierarchical neural networks[26,36] in Fig. 1. In the optimization of ML models, the output activations from the fully connected layers in the regression module are directly passed into the theory module as a vector. Those vector elements are partitioned into different parts and assigned to the $d$-band moments of the site atoms and interaction parameters of individual adsorbate frontier orbitals with the metal $sp$- and $d$-states. The binding energy of the adsorbate and the projected density of states onto adsorbate orbitals can then be computed from the theory module. For comparison purposes, the fully connected neural network (FCNN) and crystal graph convolutional neural network (CGCNN)[26,36] models were developed using the Adaptive Moment Estimation algorithm with weight decay (AdamW), see the details of input features and model optimization in the "Methods" section. The complete code, named TinNet, is available at a Github repository https://github.com/hlxin/tinnet for public access.

**Model benchmark.** A comparison of the TinNet with the purely regression-based FCNN and CGCNN on predicting the chemical reactivity of $d$-block metal surfaces is shown in Fig. 2a. The dataset corresponds to *OH at 748 {111}-terminated transition-metal surfaces with a wide variety of site compositions. Specifically, it includes intermetallics ($A_3B$) and near-surface alloys ($A'@A_{ML}$, $A–B@A_{ML}$, $A_3B@A_{ML}$, $A@A_2B_2$, and $A@AB_3$), where A (or A') represents 10 fcc/hcp metals and B covers 26 $d$-metals across the periodic table, see the "Methods" section for

computational details. OH is adsorbed at the atop site while the O−H bond is tilted toward the bridge site. The straight-up *OH adsorption configuration is less favorable than the tilted ones on transition metals because of the directional $1\pi$-orbital interactions with metal $d$-states. In this study, we did not include other local minima of tilted *OH adsorption configurations. In the feature representation, bonding angles are also not included in the CGCNN/TinNet framework. Note that other developments that are built upon the CGCNN, e.g., iCGCNN[47], and ALIGNN[48], have implemented angle features, which will be useful if multiple local minima exist in the dataset. Compared with previous studies that include different surface terminations and adsorption sites[17,26], we are focusing on a relatively small but representative dataset[14,49,50]. For *OH, we explicitly included the $3\sigma$, $1\pi$, and $4\sigma^{*}$ frontier molecular orbitals in the network design. To rigorously evaluate the prediction performance of ML models with a balanced bias/variance trade-off, we adopted $k$-fold cross-validation ($k = 10$) to optimize hyperparameters, including learning rate, # of atomic features, # of convolution-pooling layers, # of hidden layers, and # of hidden neurons of each layer[51]. A validation set (10%) is randomly split off the training set for early stopping of the optimization process as a form of regularization to avoid overfitting. In Fig. 2a, we present the learning curves of the FCNN, CGCNN, and TinNet models, in which the

mean absolute error (MAE) of prediction and its standard deviation are estimated by the nested 10-fold cross-validation approach[52] (see Supplementary Table 1 for the hyper-parameters of each model scheme). We include a diagram of the TinNet model architecture and hyperparameters in Supplementary Fig. 2 for *OH to further clarify the flow/mapping of graph features to target properties. In the data-scarce region, the FCNN showed a relatively accurate and stable prediction of *OH adsorption energies compared with CGCNN and TinNet models because of employing physics-based features (e.g., orbitalwise coordination numbers[13]) rather than low-level graph features. As the number of training samples increases, the TinNet can attain a 0.118 eV MAE of prediction with a 0.022 eV deviation, outperforming the FCNN ($0.152 \pm 0.015$ eV) and on par with the CGCNN ($0.114 \pm 0.025$ eV). Figure 2b shows a 2D histogram representing the TinNet-predicted *OH adsorption energies of all 10-fold test sets against DFT-calculated values. In graph representation, the strain and ligand effects on site reactivity can be captured by atomic features and neighboring information. For the TinNet framework, graph representation of the local coordination environment is naturally reflected by the output activations from the regression module, including (1) the $d$-band center (1st moment) and width (2nd moment) of the site atoms and (2) interaction parameters of individual adsorbate frontier orbitals with the metal $sp$- and $d$-states, such as the orbital overlap and coupling coefficients which are dependent on $d$-orbital radii, interatomic distances, and local electron densities based on the tight-binding theory[45]. To make a clear benchmark comparison of the TinNet/CGCNN/FCNN models in this work and previously published ML models of *OH chemisorption on alloy surfaces, we have tabulated their feature type, learning algorithm, # of tuning parameters, # of samples, data range, and prediction errors (MAE and RMSE) in Table 1. In a comparison of those methods, FCNN and CGCNN models rely on data to learn the underlying correlations between a site structure and the adsorption energy of *OH in a pure regression fashion, while the TinNet embeds the well-established physics, i.e., the Newns–Anderson model within the $d$-band theory of chemisorption, into the network architecture. Compared to the Bayschem model[31] trained with pristine transition-metal data (Supplementary Fig. 7), the significant improvement of the prediction accuracy (MAEs, Bayeschem: 0.27 eV, TinNet: 0.118 eV) can be attributed to the design of the TinNet architecture, allowing the algorithms to learn local interaction parameters of individual adsorbate frontier orbitals with the metal $sp$- and $d$-states from data samples of diverse site coordination environments. In contrast to ML models with hand-crafted features[13,14,21,25,31,41], the electronic structure of test samples is not needed for prediction using the TinNet. This elaborate design of the network architecture, as seen in Fig. 1, further improves the transferability of the TinNet

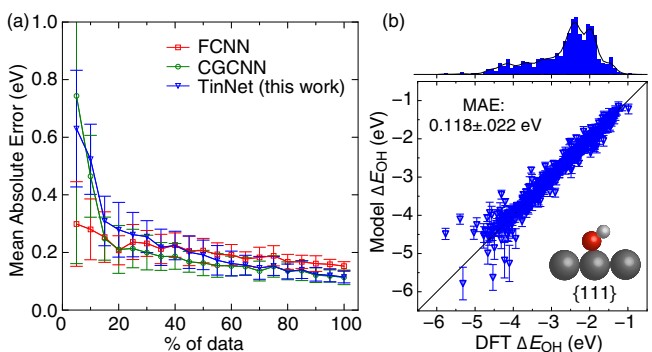

**Fig. 2 Model development. a** Learning curves of FCNN, CGCNN, and TinNet (this work) models of *OH adsorption energies on {111}-terminated intermetallics and near-surface alloys with respect to the number of available data samples. The error bar corresponds to the standard deviation of the error estimates from nested 10-fold cross-validation. **b** DFT-calculated vs. TinNet-predicted *OH adsorption energies for all 10-fold test sets, along with a histogram of data sampling. The error bar corresponds to the standard deviation of the error estimates from 10 final models. Note: MAE represents mean absolute error, the color scheme of atoms includes light gray (H), red (O), and dark gray (metal).

**Table 1 Benchmark comparison of ML models of *OH chemisorption on alloy surfaces.**

| Source | Algorithm | Representation | # of parameters | # of samples | Range (eV) | MAE (eV) | RMSE (eV) |
|---|---|---|---|---|---|---|---|
| Li et al.[14] | ANN[a] | Electronic descriptors | 106 | 635 | 1.8 | – | 0.240 |
| This work | ANN[a] | Geometric descriptors | 50,291 | 748 | 4.8 | 0.152 | 0.222 |
| Mamun et al.[20] | GPR[b] | Connectivity matrix | Nonparametric | 1235 | 4.6 | 0.170 | 0.240 |
| Bayeschem, Wang et al.[31] | Bayesian | Density of states | 11 | 512 | 2.2 | 0.160 | 0.209 |
| Bayeschem, this work | Bayesian | Density of states | 11 | 748 | 4.8 | 0.270 | 0.435 |
| DOSnet, Fung et al.[25] | CNN[c] | Density of states | 1,718,301 | 1103 | 5.4 | 0.156 | 0.221 |
| CGCNN, this work | CNN[c] | Graph | 62,593 | 748 | 4.8 | 0.114 | 0.189 |
| TinNet, this work | CNN[c] | Graph | 281,339 | 748 | 4.8 | 0.118 | 0.188 |

[a]Artificial neural network.
[b]Gaussian process regression.
[c]Convolutional neural network.

framework and signifies its potential as a robust ML approach for guiding catalyst design beyond labeled material structures.

**Model validation with single-atom alloys**. To test the prediction performance of those final models for unseen data, we chose single-atom alloys (SAAs)[53] as an out-of-sample material system that was not used in model training and cross-validation. This emerging type of material has received substantial interest in recent years because of its simplicity in the structure allowing us to control catalytic properties at the atomic level. Here, we calculated *OH adsorption at the atop site of SAAs with Cu, Ag, or Au as the host and 26 $d$-metals as the single-atom active site. Because of the limited overlap between the $d$-states wavefunction of an active $d$-metal and that of the inert host, most of those SAAs exhibit previously unseen free-atom-like $d$-states[54,55], resembling the localized electronic structure in homogeneous molecular catalysts. With the $Cu_1$/Ag(111) single-atom alloy as a specific case, recent spectroscopic measurements validated the formation of such peaky $d$-states and its effect on surface reactivity of $Cu_1$ sites[55]. Using the TinNet-predicted interaction parameters ($\Delta_0^i$, $\epsilon_a^i$, $\alpha^i$, and $\beta^i$, where $i$ represents an adsorbate frontier orbital) of $Cu_1$/Ag(111) from the regression module, Fig. 3a shows the model-constructed projected density of states onto the OH $3\sigma$, $1\pi$, and $4\sigma^*$ orbitals against with DFT-calculated distributions. The $d$-states distribution $\rho_d$ of a $Cu_1$ site and its Hilbert transform along with the adsorbate line $y = (\epsilon - \epsilon_a)/\pi\beta V_{ad}^2$ for each orbital are plotted for the graphical solution of the Newns–Anderson model[43]. The intersections in Fig. 3a represent either the adsorbate-substrate bonding and antibonding states (2 localized roots) for $1\pi$ or the resonance state (1 localized root) for $3\sigma$ and $4\sigma^*$. Given the simplicity of the model, the clearly captured strong-coupling and weak-coupling signatures for $1\pi$ and $3\sigma/4\sigma^*$ orbitals, respectively, justified the TinNet in qualitatively predicting the electronic structure of an adsorbate–substrate system. In another aspect, the comparison of model performance for predicting *OH adsorption energies between FCNN, CGCNN, and TinNet is shown in Fig. 3b and Supplementary Fig. 4. Using the 10-fold cross-validated final models, the TinNet (MAE:

0.161 ± 0.008 eV) improves its prediction error over the FCNN (MAE: 0.193 ± 0.026 eV) and CGCNN (MAE: 0.179 ± 0.029 eV), particularly for the region involving highly active early transition metals. Supplementary Fig. 5 shows the DFT-calculated vs. model-predicted $d$-band center $\epsilon_d$ and full-width $W_d$ (MAE: 0.13 and 0.37 eV, respectively) that were used to construct the semi-ellipse representing the projected $d$-states distribution $\rho_d$ onto a metal site. As an additional metric of model performance, the MAEs of the TinNet-predicted $\rho_a^i$ are 0.0205, 0.0166, and 0.0187 eV$^{-1}$ for the OH $3\sigma$, $1\pi$, and $4\sigma^*$ orbitals, respectively. To better understand the origin of the improved generalization performance, we have re-trained the FCNN and CGCNN models using multi-task learning (MTL), i.e., including both the adsorption energy and the $d$-band moments of the adsorption site in the loss function. We found that the generalization error of the adsorption energy prediction of SAAs remains similar or slightly worsens for the FCNN (MAE: 0.198 ± 0.039 eV) and CGCNN (MAE: 0.185 ± 0.029 eV). The improved generalization performance can be attributed to the solid physical basis of the TinNet framework for property prediction of out-of-sample systems with unseen structural and electronic features, rather than accessing more electronic structure information. It is important to note that optimizing hyperparameters in deep learning architectures and training deployable models with a rigorous validation procedure is quite expensive even with current GPU architectures ($10^2–10^3$ GPU hours). Future development of the TinNet framework should enable transfer learning of trained model parameters to other adsorbate systems. For adsorbates with an identical set of frontier orbitals, e.g., atomic $p_x$, $p_y$, and $p_z$ orbitals of C, N, and O adatoms, it is natural to start from past fittings since the output vectors from the regression module have the same length and physical meaning of individual adsorbate frontier orbital interacting with the metal $sp$- and $d$-states. For adsorbates with a distinct set of frontier orbitals, e.g., O, OH, and OOH, it is generally accepted that the underlying physics or factors governing the interaction strength of those adsorbates with alloy surfaces are universal. In that scenario, convolution filter parameters that extract high-level feature representations of adsorption sites can be preloaded to speed up optimization processes.

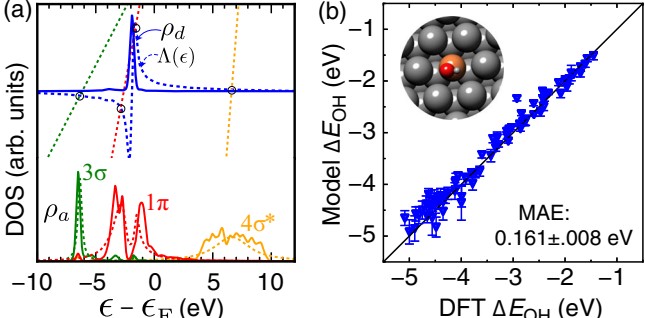

**Fig. 3 Out-of-sample validation of the TinNet model. a** Projected density of states $\rho_a$ onto the OH $3\sigma$, $1\pi$, and $4\sigma^*$ orbitals from DFT calculations (solid) and TinNet models (dashed), taking $Cu_1$/Ag(111) as an example (Ag: dark gray, Cu: brown). The graphical solution to the Newns–Anderson model is also shown, in which the intersections of the adsorbate line $y = (\epsilon - \epsilon_a)/\pi\beta V_{ad}^2$ for each orbital with the Hilbert transform $\Lambda(\epsilon)$ of the density of $d$-states $\rho_d$ represent the adsorbate–substrate bonding and antibonding states (2 localized roots) for $1\pi$ and the resonance state (1 localized root) for $3\sigma$ and $4\sigma^*$. **b** DFT-calculated vs. TinNet-predicted *OH adsorption energies for out-of-sample single-atom alloys. A broad range of transition-metal atoms (26 in total) were used as the single-site substitute of the coinage metal host, i.e., Cu, Ag, and Au. The error bar corresponds to the standard deviation of the error estimates from 10 final models.

## Discussion

A significant advantage of the TinNet framework is the model interpretability empowered by the theory module. To provide physical insights into the reactivity trend of *OH at transition-metal surfaces, we deconvolute the $d$-contributed adsorption energy $\Delta E_d$ into Pauli repulsion and orbital hybridization (see Fig. 4a). Not surprisingly, orbital hybridization dominates the overall trend of *OH adsorption energies, in agreement with the Bayesian chemisorption model developed for pure metals[31]. In the strong-binding region, the Pauli repulsion due to orbital orthogonalization involving less than half-filled $d$-shells is expected to be negligible, very well captured by the TinNet. However, it becomes prominently important for late transition metals with completely or nearly filled $d$-states[3,33]. Although this phenomenon was recognized, leveraging this physical aspect of chemical bonding for catalyst design in addition to strain[5] and ligand[4] effects has not been realized. For the diverse sites considered here, neither the $d$-band center nor the upper edge is linearly correlated with the *OH adsorption energy ($R^2$: 0.64 and 0.49, respectively) (see Supplementary Fig. 6). We argue that a linear descriptor of this kind might not exist for such a diverse dataset. Interestingly, the TinNet-predicted coupling integral squared $V^2$, i.e., $\beta V_{ad}^2$, correlates very well with the orbital hybridization energies for $3\sigma$ ($R^2 \sim 0.93$), $1\pi$ ($R^2 \sim 0.87$), and $4\sigma^*$ ($R^2 \sim 0.89$) orbitals (see Fig. 4b). This result showcases the ability

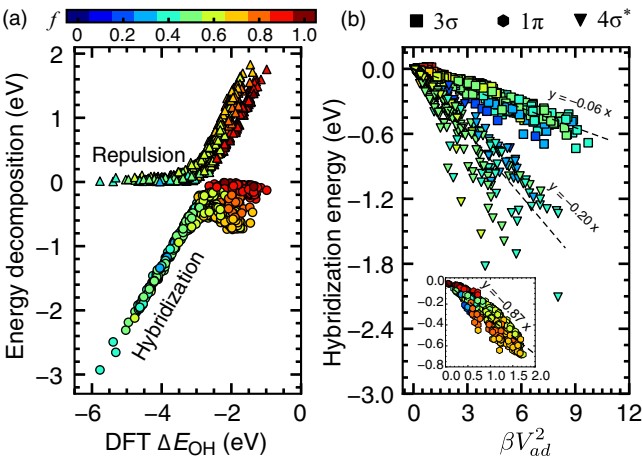

**Fig. 4 Physical insights into chemical bonding. a** Orbital hybridization and Pauli repulsion contributions from the metal $d$-states to the *OH adsorption energies on all 10-fold test sets deconvoluted by the TinNet models. **b** The TinNet-predicted coupling integral squared $\beta V_{ad}^2$ for $3\sigma$, $1\pi$, and $4\sigma^*$ orbitals linearly correlate with the corresponding orbital hybridization energy ($R^2$: 0.93, 0.87, and 0.89, respectively). Regression lines with the intercept at 0 are shown. To avoid overlap, the $1\pi$ data are plotted in the inset. All markers are color-coded according to the theoretical $d$-band filling $f$ of the *OH adsorption site.

of the TinNet framework to provide a detailed physical interpretation of the reactivity trend of metal sites that is inaccessible with purely regression-based models.

To demonstrate the approach for other adsorbates and facets, we developed the TinNet models for *O at the atop the site of the {111}-terminated bimetallic alloy surfaces and *N at the hollow site of {100}-terminated ternary alloy surfaces, as shown in Fig. 5. The 10-fold cross-validated MAEs are 0.147 and 0.116 eV for *O and *N, respectively. We use the same set of alloy surfaces for *O as the *OH models (748 total). For *N adsorbed at the four-fold hollow site, we used 329 {100}-terminated Pt-based ternary alloy surfaces ($Pt_3M$ and $Pt_2M_2$ intermetallics with M′ dopants at different positions of the top two layers, see the "Methods" section for details). *N adsorption at metal sites represents an important reactivity descriptor for ammonia electro-oxidation as the anode reaction in direct ammonia fuel cells[56–58]. We note that the surface has a coadsorbed *OH spectator species for all the samples. Our previous study has shown that *OH play a crucial role in stabilizing $*NH_x$ species under relevant operating conditions[59]. The dataset showcases the inclusion of adsorbate–adsorbate interactions in developing ML models. In the current TinNet implementation, for an $n$-atom site ensemble, the regression module automatically allocates $2n$ output neurons for the 1st and 2nd moments of the $d$-states distribution of site atoms. The $d$-states distribution of the adsorption site will be represented by a superposition of individual $d$-dos constructs, e.g., semi-elliptic functions. Other output neurons representing interaction parameters of the adsorbate frontier orbitals with the metal $sp$- and $d$-states have the same dimension and physical meanings for adsorption sites of different atom ensembles.

This study highlights the importance of the frontier molecular orbital theory, electronic structure methods, and deep learning algorithms in developing interpretable ML models of chemical bonding. Infusing theory into ML fueled with ab initio adsorption properties will eventually lead us to better understand the fundamentals of linear energy relationships[60,61] and devise strategies to overcome such constraints in catalysis[62]. For example, electrolyte molecules or ions can exert an additional coupling term

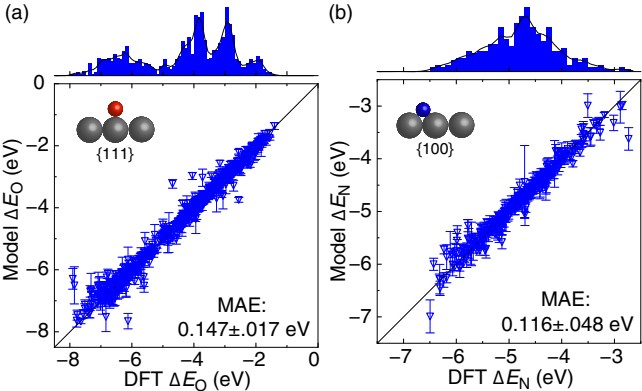

**Fig. 5 TinNet models for other adsorbates/facets.** DFT-calculated vs. TinNet-predicted (**a**) *O adsorption energies at the atop the site of {111}-terminated alloy surfaces and (**b**) *N adsorption energies at the hollow site of {100}-terminated alloy surfaces for all 10-fold test sets, along with a histogram of data sampling. *Note:* the color scheme of atoms includes red (O), blue (N), and dark gray (metal). The error bar corresponds to the standard deviation of the error estimates from 10 final models.

with the adsorbate energy level $\epsilon_a$, often via hydrogen bonding[63,64], which could be leveraged to break the adsorption-energy scaling relations for hydrogen-containing species. Indeed, there is evidence that adding a co-solvent or ionic species into the bulk electrolyte does have a positive effect on stabilizing charge-transfer intermediates in metal-air batteries[65], ammonia synthesis[66], $CO_2$ reduction[67], and oxygen evolution[68]. This physical aspect of chemical bonding can be built into the TinNet for screening improved catalytic systems with consideration of electrolyte choices. As a related note, all the structures used in this study are DFT-optimized local minima. Informing the learning algorithms of this physical information (forces are less than a threshold) in the spirit of incorporating physics, if the forces are accessible in the TinNet framework, can further constrain deep learning models and improve their transferability. Beyond a better estimation of adsorption energetics that is extensively explored in the field of catalysis, activation barriers, adsorbate–adsorbate interactions, and surface segregation energies are also important for predicting reaction kinetics and site stability prior to catalyst screening. The framework proposed here is a step toward that direction.

To conclude, the herein proposed theory-infused neural network (TinNet) represents a generalized ML approach to predicting the chemical reactivity of solid surfaces with atomically tailored active sites. Importantly, physical insights by learning from data come naturally with the TinNet, which cannot be obtained otherwise using purely regression-based methods, irrespective of feature representations. We demonstrate the approach using simple adsorbates (e.g., *OH, *O, and *N) at active site ensembles as specific cases, and it can also be transferred directly to other descriptor species and nanostructures of different site geometries or electronic complexities, e.g., metal compounds with strongly correlated $d$ electrons, paving the path toward interpretable ML discovery of novel motifs with desired catalytic properties. This study encapsulates all of the important ingredients of the ML approach and can be straightforwardly extended to generic models or principles where the neuron representing parameters should be treated on a case-by-case basis.

## Methods
**DFT calculations.** Spin-polarized DFT calculations of *OH and *O adsorption systems were performed through Quantum ESPRESSO[69] with ultrasoft

pseudopotentials. The exchange-correlation was approximated within the generalized gradient approximation (GGA) with Perdew–Burke–Ernzerhof (PBE)[70]. {111}-terminated metal surfaces were simulated using ($2 \times 2$) supercells with 4 layers and a vacuum of 15 Å between two images. The bottom two layers were fixed while the top two layers and adsorbates were allowed to relax until a force criteria of 0.1 eV/Å. A plane-wave energy cutoff of 500 eV was used. The *N adsorption systems consist of {100}-terminated Pt-based bimetallic surfaces doped with a third element. It includes Pt$_3$M and PtM bimetallics where M can be any of the transition metals, while the dopants cover 15 elements: Fe, Zn, Cu, Co, Ni, Rh, Pd, Ag, Ir, Pt, Au, Ru, Mo, Cr, and W. Spin-polarized DFT calculations were performed through Vienna ab initio simulation package (VASP) with projector-augmented wave pseudopotentials. The exchange-correlation was approximated within the GGA with the revised Perdew–Burke–Ernzerhof (RPBE)[71]. A plane-wave energy cutoff of 450 eV was used. The {100}-terminated alloy surfaces were modeled using ($2 \times 2$) supercells with 4 layers and a vacuum of 15 Å between two images. The bottom two layers were fixed while the top two layers and adsorbates were allowed to relax until force criteria of 0.05 eV/Å. In order to consider the effect of aqueous solvation on adsorption energies, an implicit solvation model was employed through the VASPsol package[72]. All of the Pt-based alloy surfaces have coadsorbed *OH ($\theta_{OH} = 1/4$ ML) as a spectator species. Doping is simulated by replacing one of the top two-layer metal atoms with dopant metals. For both {111} and {100} terminations, a Monkhorst–Pack mesh of $6 \times 6 \times 1$ was used to sample the Brillouin zone, while for molecules and radicals only the Gamma point was used. Methfessel–Paxton smearing scheme was used with a smearing parameter of 0.1 eV for adsorbate systems and 0.001 eV for molecules. Electronic energies are extrapolated to $k_B T = 0$ eV. The projected atomic and molecular density of states were obtained by projecting the eigenvectors of the full system at a denser $k$-point sampling ($12 \times 12 \times 1$) with an energy spacing of 0.01 eV onto the ones of the part, as determined by gas-phase calculations.

**FCNN models.** A FCNN is the simplest artificial neural network, and there is no cycle between node connections. The input features of FCNN include atomic features, surface features, and bulk features, which represent characteristics of the adsorption site, the environment of the adsorption site, and properties of the entire crystal. The "BulkFingerprintGenerator.bulk_average" module of the CatLearn package[37] is used to extract properties of the adsorption site, the first two surface layers, and the bulk as atomic, surface, and bulk features, respectively. All missing properties in the module are set to zero. In addition to previous properties, atomic features also contain Pauling electronegativity ($\chi_0$), $V_{ad}^2$, and atomic radius ($r_0$) while surface features include local Pauling electronegativity ($\chi$) and orbitalwise coordination numbers (CN$^s$ and CN$^d$)[40].

**Hyperparameter optimization.** In this study, five hyperparameters, namely learning rate (lr), number of hidden layers ($n\_h$), number of neurons of each hidden layer ($h\_fea\_len$), number of convolutional layers ($n\_conv$), and the length of atomic features into the convolution ($atom\_fea\_len$), were tuned by using the random search algorithm through the Ray package[51]. lr is randomly sampled from 0.0001 to 1 with log uniform distribution. $atom\_fea\_len$, $n\_conv$, $h\_fea\_len$, and $n\_h$ are random integers between 16–112, 1–10, 32–224, and 1–10, respectively. For each model, 150 randomly selected combinations are used as the hyperparameter set for the training. For each hyperparameter set, regular 10-fold cross-validation (CV) is applied. The data set is divided into 10 folds first. A fold is used as the test set for each calculation. The rest of 90% of data set will be divided into 10 folds again and a randomly chosen one fold is used as the validation set for early stopping the training procedure. Supplementary Fig. 1 illustrates the hyperparameter optimization procedure. AdamW optimization algorithm, MSE loss function, and Softplus, Sigmoid, and ReLU activation functions are implemented in the training. Batch size and weight decay are 64 and 0.0001, respectively. If no better validation loss within 1000 epochs, the model with minimal validation loss will be selected as the final model of that fold. For FCNN and CGCNN, the loss function only contains MSE($\Delta E$), but, for TinNet, the loss function is constructed with MSE($\Delta E$) + MSE($\mu_1$) + MSE($2\sqrt{\mu_2}$) + $\lambda$[MSE($\rho_{3\sigma}$) + MSE($\rho_{1\pi}$) + MSE($\rho_{4\sigma^*}$)]. The energy contribution from the $sp$-electrons ($\Delta E_0$) and the weight of density of states ($\lambda$) are set at $-2.69$ eV and 0.01, respectively, as derived from Bayesian learning models[31]. The final loss (average 10 test loss) of that hyperparameter set will be obtained. An optimized hyperparameter set with a minimal loss for each algorithm is shown in Supplementary Table 1. These hyperparameter sets will be used for all later ML optimization. Details of the CGCNN model set can be found in refs. [22,36].

**Learning curve.** The nested 10-fold cross-validation with different proportions of the dataset (from 5% to 100% with 5% as the interval) was used to evaluate the model performance. For each proportion, the dataset is divided into 10 folds. One of the folds is used as the test set, the other fold is used as the validation set, and all other eight folds are used as the training set. Supplementary Fig. 3 illustrates the procedure for generating the learning curve with the nested 10-fold cross-validation approach. 90 models, whose test set is not equal to the validation set, are used to evaluate model performance. Those 10 models whose test set is also the validation set will be used as final models for predicting unknown systems. For

different methods, the average wall-time consumed to train a model for a given data split is shown in Supplementary Table 2.

## Data availability

The training and test data of transition-metal alloy surfaces data used in this study are available in the Github repository (https://github.com/hlxin/tinnet).

## Code availability

TinNet: https://github.com/hlxin/tinnet
 CGCNN: https://github.com/txie-93/cgcnn
 CGCNN: https://github.com/ulissigroup/cgcnn
 Ray Tune: https://docs.ray.io/en/latest/tune/index.html

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

## Acknowledgements
S.H.W., H.S.P., S.W., L.E.K.A. and H.X. acknowledge the partial financial support from the NSF CAREER program (CBET-1845531). The computational resource used in this work is provided by the advanced research computing at Virginia Polytechnic Institute and State University.

## Author contributions
L.E.K.A. and H.X. supervised the research. S.-H.W., H.S.P., S.W., and H.X. conceived the idea and designed the general approach. S.-H.W., H.S.P., and S.W. conducted DFT calculations and coding. S.-H.W. and H.S.P. performed a detailed analysis. All authors revised the manuscript.

## Competing interests
The authors declare no competing interests.
