## [Peer Review File · Nature Communications]

Infusing theory into deep learning for interpretable reactivity predictionREVIEWER COMMENTS

Reviewer #1 (Remarks to the Author):

The authors have done something quite unique produced a Machine Learning paper on chemisorption that I liked and actually believe might have great value to the scientific community (not just catalysis as metal/surface adsorbate interactions have much wider applicability). What distinguishes this model, from other studies of its kind, is that it is a hybrid ML/physics based model which at its heart is an Anderson-Newns model of electronic structure/binding which is trained to model chemisorption on metal surfaces. Note surprising when one used a hybrid model the transferability is increased further away from the training set! I have advocated for incorporation of frontier molecular orbital concepts into chemisorption models of this type in the past and the authors have successfully achieved this in a way that is intellectually pleasing, scientifically sound and convincing.

I do however have a few small suggestions the authors should consider mentioning as a note of caution to the reader:

1, parameter vs data set size. With all deep learning methods, I worry about number of parameters vs data set size. If I take figure 1 and assume 3 layers with the same number of neurons as depicted and ~800 structures (each bring ~10 pieces of information) I am not sure the input data to fitting parameter ratio is very comforting! Obviously, this is an exaggeration, but I would like the authors to disclose this issue up front and more comforting because this method will need to be trained for each new system and not everybody will be as careful as the authors! So general guidelines on this point are in order if others are to use it.

2, The model is demonstrated for OH on metals at 1/8 ML coverage but will obviously have to be retrained for each new species and probably new coverage (this should be stated in thought it is obvious), how much of the information from past fittings can be reused for the next fitting, since this is a physics based model some of these learned parameters may be extensible to the next problem and minimally provide a descent starting point.

This is really a nice paper, which I enjoyed reading, and will make an excellent addition to nature comm.

Reviewer #2 (Remarks to the Author):

I have reviewed the manuscript in which the authors reported a theory-infused neural network approach for predicting surface adsorption on transition metal surfaces. In this approach, a regression module was first included to output vectors for the sequential theory module, which further predicts the adsorption energy based on orbital overlap as in the News-Anderson Hamiltonian-based methods. To demonstrate the application of this approach, the authors used OH adsorption as a representative example and showed that the results can be interpretable, which is normally challenging for pure regression approach. They showed the results could be comparable with or better than the fully-connected neural network and graph convolution neural network. They further used the example of OH adsorption on single atom alloys to demonstrate the general capability of this approach. The novelty of this work is not the fundamental knowledge of molecular adsorption, which is well documented in literature; instead, this work is emphasized on development of an interpretable approach for predicting adsorption of surface species. Overall, I think the work is valuable and nicely described. I have a few suggestions for the authors to consider.

1. I kind of understand what each module (the regression one and the theory one) does, but it is not clearly how these two modules are interfaced. For example, I think the authors should provide a more detailed description of the flow/mapping from the regression model to the theory model.

2. The work chose a simpler problem of OH adsorption. Due to its radical nature, its adsorption configuration is rather limited. It will be interesting to know, at least through a brief description, how species that can interact with multiple surface atoms can be predicted using this approach.

This is also relevant to species interacting with a step of the metal cluster where the properties of the metal atoms are inhomogeneous.

3. It will also be interesting to know how mechanical strain effect can be built in the model. I think it can be done in the regression module, but it is not clear how it can be represented in the theory module.

Reviewer #3 (Remarks to the Author):

Nat Communications

Infusing Theory into Deep Learning for Interpretable Reactivity Prediction

Shih-Han Wang et al.

Wang et al. present a neural network model for computing the energetics of adsorbates, in particular, OH*, on transition metal (alloy) surfaces. The neural network implicitly infuses domain physics, in particular the d-band theory (with Newns Anderson type Hamiltonians) whereby the output of the convolutional followed by the fully connected layers are the terms of the chemisorption model, such as the d-band moments, coefficients for the coupling and overlap integrals, etc. which can then be used to compute the binding energy. The authors show that the resulting model has excellent out-of-sample performance vis-à-vis purely regression-based neural networks, underlying the importance of physics. The model is also able to provide accurate information about hybridization, Pauli repulsion, etc. This work corresponds to perhaps the first substantive physics-informed neural network for adsorption energies of small adsorbates on catalytic surfaces. As such, this work is timely and significant and hence publishable in this journal, however, after addressing the following questions.

1. The training data needs a little more explanation. The authors note that they have employed 700 – 800 surfaces and presumably these are minima; however, it is unclear if information pertaining to the orientation of the OH bond is distinguished in these training data (especially if there are multiple distinct local minima related to the rotation around an axis through the oxygen atom)?

2. From what this reviewer understands, inter-atom distances are used in the featurization of the graph convolutional part, therefore, a pertinent question is whether the authors used the additional information that these structures are local minima (forces are less than a threshold) in the spirit of incorporating physics?

3. The biggest criticism of this reviewer is that confining this work to OH alone does not seem to portray the power of the approach well. Showing evidence of the performance of this approach on other facets and adsorbates (preferably larger, such as bifunctional intermediates such as HCOO) will be more convincing.

4. The authors should put this work in the context of their previous work on Bayesian models for chemisorption, i.e., how does this expand on their previous work and how significant is the improvement?

Reviewer #1 (Remarks to the Author):

The authors have done something quite unique produced a Machine Learning paper on chemisorption that I liked and actually believe might have great value to the scientific community (not just catalysis as metal/surface adsorbate interactions have much wider applicability). What distinguishes this model, from other studies of its kind, is that it is a hybrid ML/physics based model which at its heart is an Anderson-Newns model of electronic structure/binding which is trained to model chemisorption on metal surfaces. Note surprising when one used a hybrid model the transferability is increased further away from the training set! I have advocated for incorporation of frontier molecular orbital concepts into chemisorption models of this type in the past and the authors have successfully achieved this in a way that is intellectually pleasing, scientifically sound and convincing.

Response: We thank the reviewer for the positive comments of the hybrid ML/physics approach to probing chemisorption phenomena at metal surfaces and its implications to a broader research community beyond catalysis.

I do however have a few small suggestions the authors should consider mentioning as a note of caution to the reader:

1, parameter vs data set size. With all deep learning methods, I worry about number of parameters vs data set size. If I take figure 1 and assume 3 layers with the same number of neurons as depicted and ~800 structures (each bring ~10 pieces of information) I am not sure the input data to fitting parameter ratio is very comforting! Obviously, this is an exaggeration, but I would like the authors to disclose this issue up front and more comforting because this method will need to be trained for each new system and not everybody will be as careful as the authors! So general guidelines on this point are in order if others are to use it.

Response: We share the same concerns with the reviewer for any deep learning models, particularly for those trained with limited data. To prevent overfitting, we employed a rigorous cross-validation procedure (S. Varma, R. Simon, Bias in error estimation when using cross-validation for model selection. BMC Bioinformatics. 7, 91, 2006), i.e., regular 10-fold cross-validation for optimizing hyper-parameters and nested 10-fold cross-validation for evaluating model performance. To make a clear benchmark comparison of the TinNet/GCNN/FCNN models in this work and some of the previously published ML models of *OH chemisorption on alloy surfaces, we have tabulated their feature type, learning algorithm, # of tuning parameters, # of samples, data range, and prediction errors (MAE and RMSE) in Table 1.

Changes: We added the Table 1 in the revised manuscript, providing a clear benchmark comparison of ML models of *OH chemisorption on alloy surfaces.

Table I. **Benchmark comparison of ML models of *OH chemisorption on alloy surfaces.**

Source	Algorithm	Representation	# of parameters	# of samples	Range (eV)	MAE (eV)	RMSE (eV)
Li et al.[14]	ANN ^a	Electronic descriptors	106	635	1.8	-	0.240
This work	ANN ^a	Geometric descriptors	50,291	748	4.8	0.152	0.222
Mamun et al.[20]	GPR ^b	Connectivity matrix	nonparametric	1,235	4.6	0.170	0.240
Bayeschem, Wang et al.[31]	Bayesian	Density of states	11	512	2.2	0.160	0.209
Bayeschem, this work	Bayesian	Density of states	11	748	4.8	0.270	0.435
DOSnet, Fung et al.[25]	CNN ^c	Density of states	1,718,301	1,103	5.4	0.156	0.221
CGCNN, this work	CNN ^c	Graph	62,593	748	4.8	0.114	0.189
TinNet, this work	CNN ^c	Graph	281,339	748	4.8	0.118	0.188

^aArtificial neural network. ^bGaussian process regression. ^cConvolutional neural network

We also added the following in the main text: “To make a clear benchmark comparison of the TinNet/CGCNN/FCNN models in this work and some of the previously published ML models of *OH chemisorption on alloy surfaces, we have tabulated their feature type, learning algorithm, # of tuning parameters, # of samples, data range, and prediction errors (MAE and RMSE) in Table 1. ”

2, The model is demonstrated for OH on metals at 1/8 ML coverage but will obviously have to be retrained for each new species and probably new coverage (this should be stated in thought it is obvious), how much of the information from past fittings can be reused for the next fitting, since this is a physics based model some of these learned parameters may be extensible to the next problem and minimally provide a descent starting point.

Response: The reviewer brought up a practical point, i.e., how to transfer previously trained TinNet models to other adsorbate systems. It is important because optimizing hyper-parameters in deep learning architectures and training deployable models with a rigorous validation procedure is quite expensive even with current GPU architectures (10^2 - 10^3 GPU hours). There are two aspects that we want to elaborate on. First, for adsorbates with an identical set of frontier orbitals, e.g., atomic p_x , p_y , and p_z orbitals for C, N, and O adatoms, it is natural to start from past fittings since the output vectors from the regression module have the same length and physical meaning of individual adsorbate frontier orbitals interacting with the metal sp - and d -states. Second, for adsorbates with a distinct set of frontier orbitals, e.g., O, OH, and OOH, it is generally accepted that the underlying physics or factors governing the interaction strength of those adsorbates with alloy surfaces are universal. In that scenario, convolution filter parameters that extract high-level feature representations of adsorption sites can be preloaded to speed up optimization processes. Those two schemes are currently under development in the TinNet package.

Changes: We added the following in the main text: “It is important to note that optimizing hyper-parameters in deep learning architectures and training deployable models with a rigorous validation procedure is quite expensive even with current GPU architectures (10^2 - 10^3 GPU hours). Future development of the TinNet framework should enable transfer learning of trained model parameters to other adsorbate systems. For adsorbates with an identical set of frontier orbitals, e.g., atomic p_x , p_y , and p_z orbitals of C, N, and O adatoms, it is natural to start from past

fittings since the output vectors from the regression module have the same length and physical meaning of individual adsorbate frontier orbital interacting with the metal *sp*- and *d*-states. For adsorbates with a distinct set of frontier orbitals, e.g., O, OH, and OOH, it is generally accepted that the underlying physics or factors governing the interaction strength of those adsorbates with alloy surfaces are universal. In that scenario, convolution filter parameters that extract high-level feature representations of adsorption sites can be preloaded to speed up optimization processes.”.

This is really a nice paper, which I enjoyed reading, and will make an excellent addition to nature comm.

Reviewer #2 (Remarks to the Author):

I have reviewed the manuscript in which the authors reported a theory-infused neural network approach for predicting surface adsorption on transition metal surfaces. In this approach, a regression module was first included to output vectors for the sequential theory module, which further predicts the adsorption energy based on orbital overlap as in the News-Anderson Hamiltonian-based methods. To demonstrate the application of this approach, the authors used OH adsorption as a representative example and showed that the results can be interpretable, which is normally challenging for pure regression approach. They showed the results could be comparable with or better than the fully-connected neural network and graph convolution neural network. They further used the example of OH adsorption on single atom alloys to demonstrate the general capability of this approach. The novelty of this work is not the fundamental knowledge of molecular adsorption, which is well documented in literature; instead, this work is emphasized on development of an interpretable approach for predicting adsorption of surface species. Overall, I think the work is valuable and nicely described. I have a few suggestions for the authors to consider.

1. I kind of understand what each module (the regression one and the theory one) does, but it is not clearly how these two modules are interfaced. For example, I think the authors should provide a more detailed description of the flow/mapping from the regression model to the theory model.

Response: We thank the reviewer for pointing out the issue of a lack of adequate description about how the regression and theory modules are interfaced. The output activations from the fully-connected layers in the regression module are directly passed into the theory module as a vector. Those vector elements are partitioned into different parts and assigned to the *d*-band moments of the site atoms and interaction parameters of individual adsorbate frontier orbitals with the metal *sp*- and *d*-states. The binding energy of the adsorbate and the projected density of states onto adsorbate orbitals can then be computed from the theory module of the TinNet package, specifically using the News-Anderson model function in the Chemisorption class.

Changes: We added the following into the main text: “In optimization of ML models, the output activations from the fully-connected layers in the regression module are directly passed into the theory module as a vector. Those vector elements are partitioned into different parts and assigned to the d -band moments of the site atoms and interaction parameters of individual adsorbate frontier orbitals with the metal sp - and d -states. The binding energy of the adsorbate and the projected density of states onto adsorbate orbitals can then be computed from the theory module.”

We also include a diagram (see below) of the TinNet model architecture and hyper-parameters in the Supplementary Fig. 2 for *OH to further clarify the flow/mapping of graph features to target properties.

Supplementary Figure 2. TinNet model architecture and hyper-parameters for *OH.

2. The work chose a simpler problem of OH adsorption. Due to its radical nature, its adsorption configuration is rather limited. It will be interesting to know, at least through a brief description, how species that can interact with multiple surface atoms can be predicted using this approach.

This is also relevant to species interacting with a step of the metal cluster where the properties of the metal atoms are inhomogeneous.

Response: To demonstrate the approach for adsorbates bonded with multiple surface atoms, we have developed the TinNet models of *N adsorbed at the four-fold hollow site of {100}-terminated alloy surfaces. Please see the response to the Reviewer #3 for more details about the training data and model performance. In the current TinNet implementation, for a N-atom site ensemble, the regression module allocates 2N output neurons for the 1st and 2nd moments of the *d*-states distribution of site atoms. The overall *d*-states distribution of the adsorption site will then be represented by a superposition of individual *d*-dos constructs, e.g., semi-elliptic functions. Other output activations that represent interaction parameters of individual adsorbate frontier orbitals with the metal *sp*- and *d*-states have the same length and physical meanings for adsorption sites of different atom ensembles.

Changes: We added the following in the main text to generalize the approach to different adsorbates and facets, “To demonstrate the approach for other adsorbates and facets, we developed the TinNet models for *O at the atop site of the {111}-terminated bimetallic alloy surfaces and *N at the hollow site of {100}-terminated ternary alloy surfaces. The 10-fold cross-validated MAEs are .147 eV and .116 eV for *O and *N, respectively. We use the same set of alloy surfaces for *O as the *OH models (748 total). For *N adsorbed at the four-fold hollow site, we used 329 {100}-terminated Pt-based ternary alloy surfaces (Pt₃M and Pt₂M₂ intermetallics with M' dopants at different positions of the top two layers). *N adsorption at metal sites represents an important reactivity descriptor for ammonia electro-oxidation as the anode reaction in direct ammonia fuel cells. We note that the surface has a coadsorbed *OH spectator species for all the samples. Our previous study has shown that *OH play a crucial role in stabilizing *NH_x species under relevant operating conditions. The dataset showcases the inclusion of adsorbate-adsorbate interactions in developing machine learning models. In the current TinNet implementation, for a N-atom site ensemble, the regression module automatically allocates 2N output neurons for the first and second moments of the *d*-states distribution of site atoms. The *d*-states distribution of the adsorption site will be represented by a superposition of individual *d*-dos constructs, e.g., semi-elliptic functions. Other output neurons representing interaction parameters of the adsorbate frontier orbitals with the metal *sp*- and *d*-states have the same dimension and physical meanings for adsorption sites of different atom ensembles.”

3. It will also be interesting to know how mechanical strain effect can be built in the model. I think it can be done in the regression module, but it is not clear how it can be represented in the theory module.

Response: The reviewer raised an excellent point. The lattice strain is captured with graphical feature representations through interatomic distances. For the TinNet framework, a graph representation of strained surfaces is naturally reflected by the output activations from the regression module, including 1) the *d*-band center (1st moment) and width (2nd moment) of the site atoms and 2) interaction parameters of individual adsorbate frontier orbitals with the metal

sp- and *d*-states, such as the orbital overlap and coupling coefficients which are dependent on *d*-orbital radii, interatomic distances, and local electron densities based on the tight-binding theory (W. A. Harrison, *Physics, Electronic Structure and the Properties of Solids: The Physics of the Chemical Bond*, Dover Publications, New York, 1989).

Changes: We added the following in the main text to broadly discuss how the strain and ligand effects are captured in the TinNet models. “In graph representation, the strain and ligand effects on site reactivity can be captured by atomic features and neighboring information. For the TinNet framework, graph representation of the local coordination environment is naturally reflected by the output activations from the regression module, including 1) the *d*-band center (1st moment) and width (2nd moment) of the site atoms and 2) interaction parameters of individual adsorbate frontier orbitals with the metal *sp*- and *d*-states, such as the orbital overlap and coupling coefficients which are dependent on *d*-orbital radii, interatomic distances, and local electron densities based on the tight-binding theory.”

Reviewer #3 (Remarks to the Author):

Wang et al. present a neural network model for computing the energetics of adsorbates, in particular, OH*, on transition metal (alloy) surfaces. The neural network implicitly infuses domain physics, in particular the *d*-band theory (with Newns Anderson type Hamiltonians) whereby the output of the convolutional followed by the fully connected layers are the terms of the chemisorption model, such as the *d*-band moments, coefficients for the coupling and overlap integrals, etc. which can then be used to compute the binding energy. The authors show that the resulting model has excellent out-of-sample performance vis-à-vis purely regression-based neural networks, underlying the importance of physics. The model is also able to provide accurate information about hybridization, Pauli repulsion, etc. This work corresponds to perhaps the first substantive physics-informed neural network for adsorption energies of small adsorbates on catalytic surfaces. As such, this work is timely and significant and hence publishable in this journal, however, after addressing the following questions.

1. The training data needs a little more explanation. The authors note that they have employed 700 – 800 surfaces and presumably these are minima; however, it is unclear if information pertaining to the orientation of the OH bond is distinguished in these training data (especially if there are multiple distinct local minima related to the rotation around an axis through the oxygen atom)?

Response: All the 748 *OH adsorption energy data are from the DFT-optimized {111}-terminated alloy surfaces (minima). It includes intermetallics (A₃B) and near-surface alloys (A'@A_{ML}, A-B@A_{ML}, A₃B@A_{ML}, A@A₂B₂, and A@AB₃), where A (or A') represents 10 fcc/hcp metals and B covers 26 *d*-metals across the periodic table. OH is adsorbed at the atop site while the O-H bond is tilted toward the bridge site. The straight-up *OH adsorption

configuration is less favorable than the tilted ones on transition metals because of the directional 1π -orbital interactions with metal d -states. In this study, we did not include other local minima of tilted *OH adsorption configurations. In the feature representation, bonding angles are also not included in the CGCNN framework (S. Back, et al., J. Phys. Chem. Lett. 10, 4401–4408, 2019.) Note that other frameworks that are built upon the CGCNN, e.g., iCGCNN (C. W. Park, C. Wolverton, Phys. Rev. Materials. 4, 063801, 2020) and ALIGNN (B. DeCost, K. Choudhary, arXiv, 2021, available at <http://arxiv.org/abs/2106.01829>), have implemented angle features, which will be useful if multiple local minima exist in the dataset.

Changes: We added the following into the main text: “OH is adsorbed at the atop site while the O-H bond is tilted toward the bridge site. The straight-up *OH adsorption configuration is less favorable than the tilted ones on transition metals because of the directional 1π -orbital interactions with metal d -states. In this study, we did not include other local minima of tilted *OH adsorption configurations. In the feature representation, bonding angles are also not included in the CGCNN framework. Note that other frameworks that are built upon the CGCNN, e.g., iCGCNN, and ALIGNN, have implemented angle features, which will be useful if multiple local minima exist in the dataset.”

2. From what this reviewer understands, inter-atom distances are used in the featurization of the graph convolutional part, therefore, a pertinent question is whether the authors used the additional information that these structures are local minima (forces are less than a threshold) in the spirit of incorporating physics?

Response: It is an excellent idea! We did not use the forces to inform the learning algorithms that the structures are actually local minima. Those physical information can further constrain deep learning models and improve their transferability. The challenge is to compute the forces in the TinNet framework. It is certainly out of the scope of this study. We anticipate its development along this direction in the community.

Changes: We added the following in the main text: “As a related note, all the structures used in this study are DFT-optimized local minima. Informing the learning algorithms of this physical information (forces are less than a threshold) in the spirit of incorporating physics, if the forces are accessible in the TinNet framework, can further constrain deep learning models and improve their transferability.”

3. The biggest criticism of this reviewer is that confining this work to OH alone does not seem to portray the power of the approach well. Showing evidence of the performance of this approach on other facets and adsorbates (preferably larger, such as bifunctional intermediates such as HCOO) will be more convincing.

Response: We thank the reviewer for the critical comment and constructive suggestions. In the revised manuscript, we include the TinNet models for *O at the atop site of the $\{111\}$ -terminated alloy surfaces and *N at the hollow site of $\{100\}$ -terminated alloy surfaces. The 10-fold cross-validated MAEs are .147 eV and .116 eV for *O and *N, respectively. We use the same set of

alloy surfaces for *O as the *OH models (748 total). For *N adsorption at the hollow site, we used 329 {100}-terminated Pt-based ternary alloy surfaces (Pt₃M and Pt₂M₂ intermetallics with M' dopants at different positions of the top two layers). *N adsorption at metal sites represents an important reactivity descriptor for ammonia electro-oxidation as the anode reaction in direct ammonia fuel cells. Pt₃Ir and PtIr alloys are the state-of-the-art catalysts for this reaction. There is a strong driving force for developing Ir-free, stable alloy catalysts with reduced overpotentials. We note that the surface has a coadsorbed *OH spectator species for all the surfaces. Our previous study has shown that *OH play a crucial role in stabilizing *NH_x species under relevant operating conditions (H. S. Pillai, H. Xin, New Insights into Electrochemical Ammonia Oxidation on Pt(100) from First Principles. Ind. Eng. Chem. Res. 58, 10819-10828, 2019). The dataset showcases the inclusion of adsorbate-adsorbate interactions in developing machine learning models. For *O at the atop site, we used two neurons to represent the *d*-band moments of the adsorption site in the same way as *OH adsorption at the atop site. For the *N at the hollow site ensemble, the regression module of the TinNet package allocates 8 output neurons for the *d*-band moments of 4 directly bonded metal atoms. The overall *d*-states distribution of the hollow site ensemble is represented by a superposition of four individual *d*-dos constructs, e.g., semi-elliptic functions. For both *O and *N, we included atomic p_x, p_y, and p_z orbitals in the theory module. The same approach can be directly applied to bifunctional intermediates, e.g., bidentate *HCOO.

Changes: We added the following in the main text to generalize the approach to different adsorbates and facets, “To demonstrate the approach for other adsorbates and facets, we developed the TinNet models for *O at the atop site of the {111}-terminated bimetallic alloy surfaces and *N at the hollow site of {100}-terminated ternary alloy surfaces. The 10-fold cross-validated MAEs are .147 eV and .116 eV for *O and *N, respectively. We use the same set of alloy surfaces for *O as the *OH models (748 total). For *N adsorbed at the four-fold hollow site, we used 329 {100}-terminated Pt-based ternary alloy surfaces (Pt₃M and Pt₂M₂ intermetallics with M' dopants at different positions of the top two layers). *N adsorption at metal sites represents an important reactivity descriptor for ammonia electro-oxidation as the anode reaction in direct ammonia fuel cells. We note that the surface has a coadsorbed *OH spectator species for all the samples. Our previous study has shown that *OH play a crucial role in stabilizing *NH_x species under relevant operating conditions. The dataset showcases the inclusion of adsorbate-adsorbate interactions in developing machine learning models. In the current TinNet implementation, for a N-atom site ensemble, the regression module automatically allocates 2N output neurons for the first and second moments of the *d*-states distribution of site atoms. The *d*-states distribution of the adsorption site will be represented by a superposition of individual *d*-dos constructs, e.g., semi-elliptic functions. Other output neurons representing interaction parameters of the adsorbate frontier orbitals with the metal *sp*- and *d*-states have the same dimension and physical meanings for adsorption sites of different atom ensembles.”

Figure 5. **TinNet models for other adsorbates/facets.** DFT-calculated vs. TinNet-predicted (a) *O adsorption energies at the atop site of {111}-terminated alloy surfaces and (b) *N adsorption energies at the hollow site of {100}-terminated alloy surfaces for all 10-fold test sets, along with a histogram of data sampling. The error bar corresponds to the standard deviation of the error estimates from 10-fold cross-validation.

4. The authors should put this work in the context of their previous work on Bayesian models for chemisorption, i.e., how does this expand on their previous work and how significant is the improvement?

Response: The current work is built upon the Bayschem model developed previously using Bayesian learning. The theory module in the TinNet package implements the Newns-Anderson model at the same level of complexity as the Bayschem. The Bayschem only has a handful of tuning parameters for typical adsorbates, e.g., *O, *OH, and *OOH. We optimize those parameters using ~ 10 transition metal data including adsorption energies and projected density of states onto individual adsorbate frontier orbitals. The MAE of predicted *OH adsorption energies on those 748 alloy surfaces from the Bayschem is .27 eV, see Supplementary Fig. S7. The significant improvement of the TinNet prediction accuracy (10-fold cross-validated MAE: .118 eV) can be attributed to the design of the TinNet architecture, allowing the algorithms to learn local interaction parameters of individual adsorbate frontier orbitals with the metal *sp*- and *d*-states from data samples of diverse site coordination environments..

Changes: We show the Bayschem model prediction of *OH adsorption energies for the same dataset in Supplementary Fig. S7. We also added the following in the main text: “Compared to the Bayschem model trained with pristine transition-metal data (Supplementary Fig. S7), the significant improvement of the prediction accuracy (MAEs, Bayschem: .27 eV, TinNet: .118 eV) can be attributed to the design of the TinNet architecture, allowing the algorithms to learn local interaction parameters of individual adsorbate frontier orbitals with the metal *sp*- and *d*-states from data samples of diverse site coordination environments.”

REVIEWERS' COMMENTS

Reviewer #3 (Remarks to the Author):

The authors have made substantial changes to the manuscript to address the issues raised by this reviewer. This revised manuscript can be published.